# Hepatotropic Peptides Grafted onto Maleimide-Decorated Nanoparticles: Preparation, Characterization and *In Vitro* Uptake by Human HepaRG Hepatoma Cells

**DOI:** 10.3390/polym14122447

**Published:** 2022-06-16

**Authors:** Clarisse Brossard, Manuel Vlach, Lucas Jacquet, Elise Vène, Vincent Dorcet, Pascal Loyer, Sandrine Cammas-Marion, Nicolas Lepareur

**Affiliations:** 1Univ Rennes, Ecole Nationale Supérieure de Chimie de Rennes, CNRS, ISCR, UMR 6226, ScanMAT, UMS2001, 35000 Rennes, France; clarisse.brossard@ensc-rennes.fr (C.B.); lucas.jacquet22@outlook.fr (L.J.); vincent.dorcet@uni-rennes1.fr (V.D.); 2INSERM, INRAE, Univ Rennes, Institut NUMECAN (Nutrition Metabolisms and Cancer) UMR_A 1341, UMR_S 1241, 35000 Rennes, France; manuel.vlach@agrocampus-ouest.fr (M.V.); elise.vene@chu-rennes.fr (E.V.); 3Institut Agro, INRAE, PEGASE, 35000 Rennes, France; 4Pôle Pharmacie, Service Hospitalo-Universitaire de Pharmacie, CHU Rennes, 35033 Rennes, France; 5Comprehensive Cancer Center Eugène Marquis, 35000 Rennes, France

**Keywords:** poly(benzyl malate), biocompatible NPs, post-formulation modification, hepatotropic peptides, HepaRG cells

## Abstract

We recently demonstrated the strong tropism of George Baker (GB) Virus A (GBVA10-9) and *Plasmodium circumsporozoite* protein (CPB) derived synthetic peptides towards hepatoma cells. In a first approach, these peptides were covalently bound to poly(benzyl malate) (PMLABe_73_) and poly(ethylene glycol)-*block*-PMLABe_73_ (PEG_62_-*b*-PMLABe_73_) (co)polymers, and corresponding peptide-decorated nanoparticles (NPs) were prepared by nanoprecipitation. We showed that peptide enhanced NPs internalization by hepatoma cells. In the present work, we set up a second strategy to functionalize NPs prepared from PMLABe_73_ derivates. First, maleimide-functionalized PMLABe_73_ (Mal-PMLABe_73_) and PEG_62_-*b*-PMLABe_73_ (Mal-PEG_62_-*b*-PMLABe_73_) were synthesized and corresponding NPs were prepared by nanoprecipitation. Then, peptides (GBVA10-9, CPB and their scramble controls GBVA10-9scr and CPBscr) with a thiol group were engrafted onto the NPs’ maleimide groups using the Michael addition to obtain peptide functionalized NPs by post-formulation procedure. These peptide-modified NPs varied in diameter and dispersity depending on the considered peptides and/or (co)polymers but kept their spherical shape. The peptide-functionalized NPs were more efficiently internalized by HepaRG hepatoma cells than native and maleimide-NPs with various levels relying on the peptide’s nature and the presence of PEG. We also observed important differences in internalization of NPs functionalized by the maleimide-thiol-peptide reaction compared to that of NPs prepared from peptide-functionalized PMLABe_73_ derivatives.

## 1. Introduction

According to the GLOBOCAN 2020 report published by the International Cancer Research Center, primary liver cancer is the sixth most commonly diagnosed cancer in the world, and the third leading cause of cancer-related death in 2020, with 906,000 new cases diagnosed and 830,000 deaths per year [1]. Primary liver cancers mainly include hepatocellular carcinomas (HCC), 75% to 85% of cases, and cholangiocarcinomas, 10% to 15% of cases [2,3,4]. Curative therapies, mainly based on surgical resection of small tumors, are only proposed for early stages of HCC, knowing that the majority of HCCs are usually detected at advanced or very advanced stages, for which only palliative treatments or symptomatic therapy can be proposed to the patient, respectively [5]. Therapies for intermediate to advanced stages of HCC currently include the administration of multi-kinase inhibitors such as sorafenib, the use of immunotherapies, radioembolization, radiotherapies and transarterial chemoembolization [6]. Combinations of several lines of treatments are also being evaluated in clinical trials [7,8,9,10]. However, despite undeniable progress in the management of advanced HCC, the overall survival of patients remains limited and often associated with severe side effects leading to the discontinuation of treatments.

New solutions are therefore needed for both early diagnosis and therapy of HCC. Targeting tumoral hepatocytes thanks to monoclonal antibodies, polysaccharides or peptides grafted onto biomacromolecules, nanoparticles or nanoplatforms might therefore be an efficient solution in the design of innovative and personalized nanomedicines for early diagnosis and therapy of HCC.

In this context, we have recently described the strong hepatotropism of two short synthetic peptides: the George Baker (GB) Virus A (GBVA10-9) and *Plasmodium circumsporozoite* protein (CPB) derived synthetic peptides [11]. In a first step, we have coupled these two peptides and their scrambled versions (GBVA10-9scr and CPBscr), C-terminated by the mercaptopropanoic acid (thiol group), to the maleimide end-chain of both hydrophobic PMLABe_73_ and amphiphilic PEG_62_-*b*-PMLABe_73_, an engrafting method herein referred as **pre-formulation** method allowing to produce peptide-functionalized NPs targeting hepatoma cells [12]. Corresponding peptide-decorated NPs were prepared, and in vitro assays using HepaRG human hepatoma cell lines demonstrated that there were both a peptide and a linker effects for the CPB, while it seemed that the GBVA10-9 was more probably entrapped into the NPs’ hydrophobic inner core because of the hydrophobic nature of this peptide, since no peptide effect was noticed on cell uptake [12]. Valetti et al. [13] have described this phenomenon of targeting molecule entrapment into NPs while studying the preparation of squalene-based NPs decorated with the CKAAKN peptide. They showed that the procedure used to graft peptide, before or after the formulation, has a significant influence on peptide targeting efficiency [13].

Our results and those obtained by Valetti et al. prompted us to consider alternative solution to prepare peptide-decorated NPs, and evaluate the effect of the procedure (before or after formulation) used to graft selected peptides on the in vitro uptake efficiency of peptide-decorated NPs. Several authors have grafted the targeting molecules (anti-body, peptide, etc.) onto formulated nanovectors having on their surface modifiable groups such as carboxylic acid or thiol groups [13,14]. Indeed, the selected targeting molecules have to be grafted onto the pre-formed nanovectors under mild conditions that do not lead to nanovectors destabilization and/or unwanted release of encapsulated molecules [13,14].

In this context, we studied the grafting of the selected peptides (GBVA10-9, GBVA10-9scr, CPB and CPBscr) C-terminated by a thiol group onto preformed maleimide (Mal)-exhibiting NPs, previously obtained by nanoprecipitation of a mixture of either Mal-PMALBe_73_ (10 wt%) and PMLABe_73_ (90 wt%) or Mal-PEG_62_-*b*-PMLABe_73_ (10 wt%) and PMLABe_73_ (90 wt%). This method of coupling peptides onto (co)polymers is further referred as **post-formulation** procedure. Resulting peptide-decorated NPs have been characterized by dynamic light scattering (DLS) to determine their average diameter (Dh) and dispersity (PDI), and by transmission electron microscopy (TEM) to visualize their morphology and confirm their diameter and dispersity. Finally, internalization of such peptide-decorated NPs, prepared by the post-formulation method, by HepaRG hepatoma cells was evaluated using flow cytometry. All the obtained results were compared to the ones previously observed with peptide-decorated NPs gotten by the pre-formulation procedure.

## 2. Materials and Methods

### 2.1. Materials and Apparatus

All chemicals were used as received. -maleimide, -carboxylic acid PEG_62_ (Mw = 3000 g/mol, *n* = 62) and α-methoxy, ω-carboxylic acid PEG_42_ (Mw = 2015 g/mol, *n* = 42) were purchased from Iris Biotech GmbH (Iris Biotech GmbH, Marktredwitz, Germany). Tetraethylammonium hydroxide and 6-maleimidohexanoic acid were purchased from Sigma-Aldrich (Sigma-Aldrich, Saint-Louis, MO, USA). Peptides were provided by Eurogentec (Eurogentec, Liege, Belgium). 1,1′-Dioctadecyl-3,3,3′,3′-tetramethylindo dicarbocyanine perchlorate (DiD Oil) was purchased from Invitrogen (Thermo Fisher Scientific, Illkirch Graffenstaden, France). Solvents were purchased from Sigma-Aldrich (Sigma-Aldrich, Saint Quentin Fallavier, France).

*Nuclear Magnetic Resonance spectroscopy (NMR):* The standard temperature was adjusted to 298 K. NMR spectra were recorded on a Bruker Avance III 400 spectrometer (Bruker, Wissembourg, France) operating at 400.13 MHz for ^1^H, equipped with a BBFO probe with a Z-gradient coil and a GREAT 1/10 gradient unit. The zg30 Bruker pulse program was used for 1D ^1^H NMR, with a TD of 64 k, a relaxation delay d1 = 2 s and 8 scans. The spectrum width was set to 18 ppm. Fourier transform of the acquired FID was performed without any apodization in most cases.

*Size Exclusion Chromatography**(SEC):* Weight average molar mass (Mw) and dispersity (Đ = Mw/Mn) values were measured by SEC in THF at 40 °C (flow rate = 1.0 mL/min) on a GPC2502 Viscotek apparatus equipped with a refractive index detector Viscotek VE 3580 RI, a guard column Viscotek TGuard, Org 10 × 4.6 mm, a LT5000L gel column 300 × 7.8 mm and a GPC/SEC OmniSEC Software (Malvern, Worcestershire, UK). The polymer samples were dissolved in THF (2 mg/mL). All elution curves were calibrated with polystyrene standards.

*Differential scanning calorimetry (DSC):* DSC measurements were performed on NETZSCH DSC 200 F3 instrument equipped with an intracooler (ERICH NETZSCH GMBH and CO. Holding KG, Selb, Germany). DSC traces were measured at 10 °C/min up heating (red curve) and at −5 °C/min upon cooling (purple curve). Glass transition temperature (Tg) was defined as the midpoint of the glass transition process upon the heating process.

*Dynamic Light Scattering (DLS)**:* DLS measurements were performed on a Nano-sizer ZS90 (Malvern, Worcestershire, UK) at 25 °C, with a He-Ne laser at 633 nm and a detection angle of 90 °C. Three runs of 70 scans each were performed on each NPs suspension, and average values of hydrodynamic diameter (Dh) and dispersity (PDI) were given. The size distribution reports were given by Intensity. Two examples of report were given in the SI files (Appendix A).

*Transmission electron microscopy (TEM)**:* TEM images were recorded using a Jeol 2100 microscope (Jeol, Tokyo, Japan) equipped with a Glatan Orius 200D camera using an 80 KeV accelerating voltage on the THEMIS platform (ISCR–Rennes, France). Each sample was deposited on a Formvar-carbon film coated on a 300-mesh copper grid. After 6 min, the excess of sample was removed and a staining was realized with phosphotungstic acid (1 v%).

*Flow cytometry**:* Cells were analyzed by flow cytometry using a LSRFortessa™ X-20 cytometer (Becton Dickinson, Becton Drive Lake, NJ, USA), using the cytometry core facility of the Biology and Health Federative research structure Biosit (Rennes, France) to quantify the fluorescence emitted by the DiD Oil-loaded NPs within cells. Cytometry data were analyzed using FACSDiva^TM^ software (Becton Dickinson, Becton Drive Lake, NJ, USA).

### 2.2. Methods

#### 2.2.1. Synthesis and Characterization of (co)Polymers

PMLABe_73_, PEG_42_-*b*-PMLABe_73_, Mal-PMLABe_73_ and Mal-PEG_62_-*b*-PMLABe_73_ were synthesized, as described previously [12], by anionic ring-opening polymerization (aROP) of benzyl malolactonate (MLABe) in presence of tetraethylammonium benzoate, tetraethylammonium α-methoxy,ω-propanoate PEG_42_, tetraethylammonium 6-maleimidohaxanoate and tetraethylammonium α-maleimido,ω-propanoate PEG_62_, respectively, as initiators [12]. The monomer, MLABe, was synthesized as described previously [15]. After purification by precipitation in ethanol, the (co)polymers were dried under vacuum at room temperature for 24 h and characterized by ^1^H NMR (structure and NMR molar mass, Figure 1 and Appendix A), SEC (weight average molar mass and dispersity, Figure 1 and Appendix A) and DSC (glass transition temperature, Figure 1 and Appendix A).

#### 2.2.2. NPs Formulation and Peptides’ Grafting Procedure

Maleimide-decorated and native NPs encapsulating 0.1 wt% of the fluorescence probe DiD Oil were formulated using the nanoprecipitation method as previously described by Fessi et al. [16] and peptides were grafted onto maleimide-decorated NPs using Michael addition [17].

##### Native PMLABe_73_-Based NPs (NPs 1 and NPs 1′)

*-NPs 1:* 140 µL of a solution of PMLABe_73_ at a concentration of 7.14 mg/mL in DMF were mixed with 10 µL of a DiD Oil solution in dimethylformamide (DMF) at a concentration of 0.1 mg/mL (0.1 wt%). This blue solution was rapidly added to 2 mL of distilled water under vigorous stirring and the resulting suspension was stirred at room temperature (RT) for 15 min. The bluish suspension was analyzed by DLS (Figure 2).

*-NPs 1*′: 8.5 μL of PBS pH7.4 were added to NPs 1 suspension, and the mixture was stirred for 5 min at RT. The suspension was deposited on a Sephadex column, eluted by 0.5 mL then 3.5 mL of water, and analyzed by DLS (Table 1).

##### Maleimide-Decorated PMLABe_73_-Based NPs (NPs 2 and NPs 2′)

*-NPs 2:* 5 µL of a solution of Mal-PMLABe_73_ (10 wt%) at a concentration of 20 mg/mL in DMF were mixed with 135 µL of a solution of PMLABe_73_ (90 wt%) at a concentration of 6.67 mg/mL in DMF. 10 µL of a DiD Oil solution in DMF at a concentration of 0.1 mg/mL (0.1 wt%) were then added. This blue solution was rapidly added to 2 mL of distilled water under vigorous stirring and the resulting suspension was stirred at RT for 15 min. The bluish suspension was analyzed by DLS (Figure 2).

*-NPs 2*′: 8.5 μL of PBS pH7.4 were added to NPs 2 suspension, and the mixture was stirred for 5 min at RT. The suspension was purified by filtration through a Sephadex column as described above, and analyzed by DLS (Table 1).

##### Peptide-Decorated PMLABe_73_-Based NPs (NPs 3 to NPs 6)

The Mal-decorated PMLABe_73_ NPs encapsulating DiD Oil were prepared as described for NPs 2, followed by: (*i*). for NPs 3, the addition of 8.5 μL of PBS pH7.4 solution of GBVA10-9 at a concentration of 2 mg/mL, (*ii*). for NPs 4, the addition of 8.5 μL of PBS pH7.4 solution of GBVA10-9scr at a concentration of 2 mg/mL, (*iii*). for NPs 5, the addition of 8.5 μL of PBS pH7.4 solution of CPB at a concentration of 2 mg/mL, (*iv*). for NPs 6, the addition of 8 μL of PBS pH7.4 solution of CPBscr at a concentration of 2 mg/mL. All the suspensions were purified by filtration through a Sephadex column as described above, and analyzed by DLS (Table 1) and TEM.

##### Native PEGylated PMLABe_73_-Based NPs (NPs 7 and NPs 7′)

*-NPs 7:* 5 µL of a solution of PEG_42_-*b*-PMLABe_73_ (10 wt%) at a concentration of 20 mg/mL in DMF were mixed with 135 µL of a solution of PMLABe_73_ (90 wt%) at a concentration of 6.67 mg/mL in DMF. 10 µL of a DiD Oil solution in DMF at a concentration of 0.1 mg/mL (0.1 wt%) were then added. This blue solution was rapidly added to 2 mL of distilled water under vigorous stirring and the resulting suspension was stirred at RT for 15 min. The resulting bluish suspension was analyzed by DLS (Figure 2).

*-NPs 7*′: 8.5 μL of PBS pH7.4 were added to NPs 7 suspension, and the mixture was stirred 5 min at RT. The suspension was purified by filtration through a Sephadex column as described above, and analyzed by DLS (Table 1).

##### Maleimide-Decorated PEGylated PMLABe_73_-Based NPs (NPs 8 and NPs 8′)

*-NPs 8:* 5 µL of a solution of Mal-PEG_62_-*b*-PMLABe_73_ at a concentration of 20 mg/mL in DMF were mixed with 135 µL of a solution of PMLABe_73_ at a concentration of 6.67 mg/mL in DMF. 10 µL of a DiD Oil solution in DMF at a concentration of 0.1 mg/mL (0.1 wt%) were then added. This blue solution was rapidly added to 2 mL of distilled water under vigorous stirring and the resulting suspension was stirred at RT for 15 min. The resulting bluish suspension was analyzed by DLS (Figure 2).

*-NPs 8*′: 8.5 μL of PBS pH7.4 were added and the mixture was stirred 5 min at RT. The suspension was purified by filtration through a Sephadex column as described above, and analyzed by DLS (Table 1).

##### Peptide-Decorated PEGylated PMLABe_73_-Based NPs (NPs 9 to NPs 12)

The Mal-decorated PEG_62_-*b*-PMLABe_73_ NPs encapsulating DiD Oil were prepared as described for NPs 8, followed by: (*i*). for NPs 9, the addition of 7.5 μL of PBS pH7.4 solution of GBVA10-9 at a concentration of 2 mg/mL, (*ii*). for NPs 10, the addition of 7.5 μL of PBS pH7.4 solution of GBVA10-9scr at a concentration of 2 mg/mL, (*iii*). for NPs 11, the addition of 7.5 μL of PBS pH7.4 solution of CPB at a concentration of 2 mg/mL, (*iv*). for NPs 12, the addition of 6.5 μL of PBS pH7.4 solution of CPBscr at a concentration of 2 mg/mL. All the suspensions were purified by filtration through a Sephadex column as described above, and analyzed by DLS (Table 1) and TEM.

#### 2.2.3. Cell Culture and In Vitro Uptake Assay of NPs

The human HepaRG hepatoma cells were grown in William’s E medium supplemented with 10% Fetal Calf Serum (FCS), 2% L-glutamine, 5 mg/L insulin, hydrocortisone hemisuccinate at 5 × 10^−5^ M, 100 units/mL penicillin and 100 µg/mL streptomycin. The HepaRG cells were cultured at 37 °C in a humidified atmosphere of 5% CO_2_, and the medium was renewed every 2 days.

For the NPs’ uptake assay, 10^5^ HepaRG cells were seeded in 24-well plates the day before the incubations with the different batches of NPs. The culture media of the cells were renewed with media containing NPs at a final concentration of (co)polymers of 25 µg/mL. The cells were incubated with NPs for 24 h, then were detached with trypsin-EDTA and resuspended in complete medium for flow cytometry analysis and evaluation of the NPs’ uptake by measuring the fluorescence emitted by the DiD oil loaded NPs.

## 3. Results

### 3.1. Synthesis of (co)Polymers

The four selected (co)polymers were synthesized as described previously by aROP of MLABe in presence of the corresponding tetraethylammonium carboxylate (Figure 1), either commercially available (tetraethylammonium benzoate) or synthesized by an acid/base reaction between the corresponding carboxylic acid and tetraethylammonium hydroxide [12].

**Figure 1 polymers-14-02447-f001:**
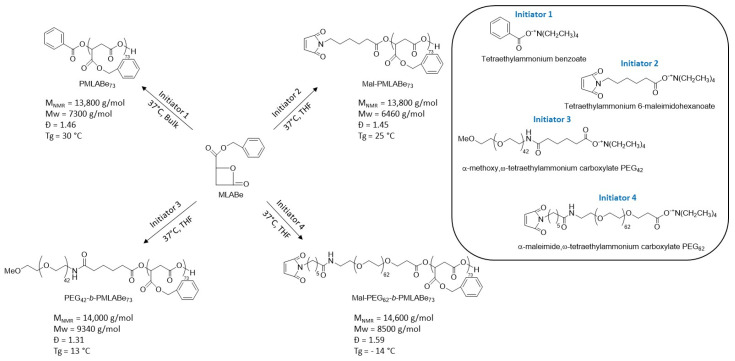
aROP of MLABe leading to PMLABe_73_, PEG_42_-*b*-PMLABe_73_, Mal-PMLABe_73_, and Mal-PEG_62_-*b*-PMLABe_73_ [12].

The molar mass of the PMLABe block has been defined by the monomer/initiator ratio and fixed at 15,000 g/mol (*n* = 73). As shown by Figure 1, two non-functionalized (co)polymers, PMLABe_73_ and PEG_42_-b-PMLABe_73_, and two maleimide-functionalized (co)polymers, Mal-PMLABe_73_ and Mal-PEG_62_-b-PMLABe_73_, have been successfully synthesized. After purification by precipitation, the proton NMR spectra (structure, Appendix A), the chromatograms obtained by size exclusion chromatography (weight average molar mass (Mw) and dispersity (Ð), Figure 1 and Appendix A) and thermograms obtained by differential scanning calorimetry (glass transition temperature, Tg, Appendix A) were recorded. The (co)polymers had the expected structures (Appendix A) and molar masses calculated from the proton NMR spectra (Appendix A) were close to theoretical ones (Figure 1). As already observed [12], weight average molar mass (Mw) measured by SEC were lower than the theoretical one and varied from 5700 to 8500 g/mol, while dispersity values were in the range of 1.45 to 1.59 (Figure 1 and Appendix A). PMLABe_73_ and Mal-PMLABe_73_ showed similar glass transition temperature at 30 °C and 25 °C, respectively (Appendix A, respectively). On the other hand, amphiphilic block copolymers, PEG_42_-b-PMLABe_73_ and Mal-PEG_62_-b-PMLABe_73_, showed also only one glass transition temperature at 13 °C and −14 °C, respectively (Appendix A, respectively).

### 3.2. Preparation and Characterization of Peptide-Decorated Nanoparticles

Native and maleimide-decorated NPs PMLABe_73_-based NPs (NPs 1 and NPs 2, respectively), and native maleimide-decorated PEGylated NPs (NPs 7 and NPs 8, respectively), containing 0.1 wt% of the fluorescence dye DiD Oil, were prepared by nanoprecipitation [16] of: (i). PMLABe_73_ (NPs 1, Figure 2A), (ii). a mixture of 90 wt% PMLABe_73_ and 10 wt% Mal-PMLABe_73_ (NPs 2, Figure 2B), (iii). a mixture of 90 wt% PMLABe_73_ and 10 wt% PEG_42_-b-PMLABe_73_ (NPs 7, Figure 2C), and (iv). a mixture of 90 wt% PMLABe_73_ and 10 wt% Mal-PEG_62_-b-PMLABe_73_ (NPs 8, Figure 2D).

**Figure 2 polymers-14-02447-f002:**
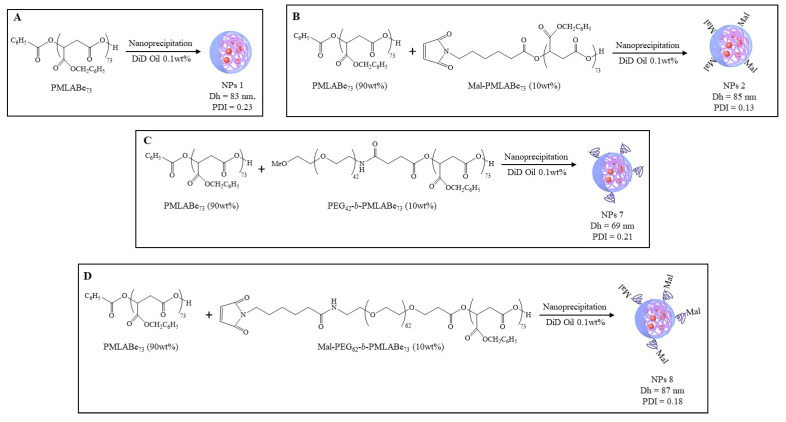
Formulation of: (**A**) Native PMLABe_73_-based NPs 1, (**B**) Maleimide-decorated PMLABe_73_/Mal-PMLABe_73_-based NPs 2, (**C**) Native PMLABe_73_/PEG_42_-*b*-PMLABe_73_-based NPs 7, (**D**) Maleimide-decorated PMLABe_73_/Mal-PEG_62_-*b*-PMLABe_73_-based NPs 8. Hydrodynamic diameter (Dh) and dispersity (PDI) were measured by DLS (3 runs of 70 scans each).

These native and maleimide-decorated NPs were analyzed by DLS to determine their hydrodynamic diameter (Dh) and dispersity (PDI). As evidenced by results gathered in Figure 2, native and maleimide-decorated NPs have relatively similar hydrodynamic diameters, varying from 69 to 87 nm, with quite low dispersity varying from 0.13 to 0.23.

Since native and maleimide-decorated NPs had the expected characteristics in terms of diameter and dispersity, we therefore proceeded to the grafting of the four selected peptides (GBVA10-9, GBVA10-9scr, CPB and CPBscr) C-terminated by a thiol group onto the preformed NPs bearing maleimide functions at their surface (Figure 3A(a,b)) using the Michael addition [17]. In parallel, control NPs 1, NPs 2, NPs 7 and NPs 8 were also treated under the same reaction conditions (PBS pH 7.4, Sephadex column) in absence of peptides (Figure 3B), thus leading to NPs 1′, NPs 2′, NPs 7′ and NPs 8′, to evaluate if these reaction conditions had any influence on NPs’ characteristics.

These peptide-free NPs and the peptide-decorated NPs were characterized by DLS (Table 1). Their hydrodynamic diameter varied from 65 to 95 nm with dispersity ranging from 0.12 to 0.26 (Table 1).

**Table 1 polymers-14-02447-t001:** Characteristics of peptide-free and peptide-decorated NPs suspensions.

Nature	Code	Composition	Dh (nm) ^a^	PDI ^a^
Peptide-free NPs	NPs 1′	PMLABe_73_	79	0.22
	NPs 2′	PMLABe_73_/Mal-PMLABe_73_ (90/10)	95	0.12
Peptide-decorated NPs	NPs 3	PMLABe_73_/GBVA10-9-PMLABe_73_ (90/10)	232	0.35
	NPs 4	PMLABe_73_/GBVA10-9scr-PMLABe_73_ (90/10)	222	0.44
	NPs 5	PMLABe_73_/CPB-PMLABe_73_ (90/10)	82	0.23
	NPs 6	PMLABe_73_/CPBscr-PMLABe_73_ (90/10)	95	0.21
Peptide-free PEGylated NPs	NPs 7′	PMLABe_73_/PEG_42_-*b*-PMLABe_73_ (90/10)	65	0.20
	NPs 8′	PMLABe_73_/Mal-PEG_62_-*b*-PMLABe_73_ (90/10)	73	0.26
Peptide-decorated PEGylated NPs	NPs 9	PMLABe_73_/GBVA10-9-PEG_62_-*b*-PMLABe_73_ (90/10)	212	0.31
	NPs 10	PMLABe_73_/GBVA10-9scr-PEG_62_-*b*-PMLABe_73_ (90/10)	159	0.37
	NPs 11	PMLABe_73_/CPB-PEG_62_-*b*-PMLABe_73_ (90/10)	75	0.21
	NPs 12	PMLABe_73_/CPBscr-PEG_62_-*b*-PMLABe_73_ (90/10)	74	0.15

^a^ Dh and PDI were measured by DLS (3 runs of 70 scans each).

On the other hand, peptides’ grafting globally led to an increase in both hydrodynamic diameters and dispersity of the corresponding NPs (NPs 3 to 6, and NPs 9 to 12—Table 1), with Dh values varying from 82 to 232 nm and PDI between 0.21 and 0.44 for non-PEGylated NPs (NPs 3 to 6—Table 1), and Dh values varying from 74 to 212 nm and PDI between 0.21 to 0.37 for PEGylated NPs (NPs 9 to 12—Table 1). We also observed that the Dh of peptide-decorated NPs formed from a mixture of PMLABe_73_ (90 wt%) and Mal-PEG_62_-*b*-PMLABe_73_ (10 wt%) were slightly smaller than those measured for their corresponding peptide-decorated NPs prepared from a mixture of PMLABe_73_ (90 wt%) and Mal-PMLABe_73_ (10 wt%).

Peptides-decorated NPs were also visualized by transmission electron microscopy (TEM, Figure 4) to evaluate the impact of peptides’ grafting on the morphology of resulting modified NPs.

TEM images (Figure 4) showed that all the peptide-modified NPs kept their spherical morphology, and that they had diameters and dispersity in agreement with those measured by DLS, knowing that the NP’s TEM images are realized under dry conditions thus usually leading to underestimated diameters.

### 3.3. Nanoparticle’s Uptake by HepaRG Hepatoma Cells In Vitro

The effect of peptide grafting on maleimide-decorated NPs by the post-formulation method onto cell uptake has been evaluated using HepaRG hepatoma cells (Figure 5). This cell line, used worldwide as an alternative to human hepatocytes [18], was also chosen in order to compare the NP’s uptake evaluated in previous studies [12]. The cells were incubated with NPs suspensions without maleimide functions or peptides on their surfaces (control NPs 1′ and NPs 7′, Table 1), with maleimide decorated NPs suspensions (control NPs 2′ and NPs 8′, Table 1) and with peptide-modified NPs (NPs 3 to 6, and NPs 9 to 12, Table 1).

As evidenced by Figure 5, NPs 1′ and NPs 7′ and maleimide-decorated NPs (NPs 2′ and NPs 8′) were poorly internalized by the HepaRG cells. Peptide-decorated NPs without PEG were significantly better internalized in a peptide-dependent manner, with higher uptake for GBVA10-9 and GBVA10-9scr decorated NPs. All NPs with PEG were less internalized demonstrating that presence of the PEG spacer reduced the cell uptake (Figure 5). However, NPs decorated with GBVA10-9 (NPs 9) or CPB (NPs 11) through the PEG spacer were significantly better internalized than the same NPs decorated with GBVA10-9scr (NPs 10) or CPBscr (NPs 12) (Figure 5).

Finally, without or with a PEG spacer, NPs decorated with GBVA10-9 (NPs 3, NPs 9) were slightly more internalized than the corresponding NPs decorated with CPB (NPs 5, NPs 11) (Figure 5).

## 4. Discussion

In the past years, our team has developed a large set of biocompatible (co)polymers derived from PMLABe, which can be used for multiple applications in the field of nanomedicine and biomaterials [19].

In this report, our first aim was to demonstrate that peptide-functionalization of NPs prepared from PMLABe derivates was possible after nanoprecipitation with a post-formulation procedure using maleimide modified PMLABe_73_ and PEG_62_-*b*-PMLABe_73_ (co)polymers. First, this study highlighted the robustness of the polymerization reaction of MLABe. Indeed, the four (co)polymers (PMLABe_73_, PEG_42_-*b*-PMALBe_73_, Mal-PMLABe_73_ and Mal-PEG_62_-*b*-PMLABe_73_) have identical structures (Appendix A), molar masses and dispersity (Figure 1 and Appendix A) to those obtained previously [12]. Furthermore, we have also completed the characterization of the (co)polymers by a DSC analysis. As expected, the two amorphous homopolymers, PMLABe_73_ and Mal-PMLABe_73_, showed only one glass transition temperature with values being in good agreement with those given by Guérin et al. [20] and Cammas et al. [21] for PMLABe homopolymers. Surprisingly, the amphiphilic block copolymers, PEG_42_-*b*-PMLABe_73_ and Mal-PEG_62_-*b*-PMLABe_73_, also showed one glass transition temperature with lower values than those observed for the PMLABe_73_ homopolymers, while one could expect to see two Tg values, one corresponding to the Tg of the PEG block and the other to the Tg of the PMLABe_73_ block. The first hypothesis that can be formulated to explain such results was that: (*i*). the Tg of PEG (Tg = −60 °C [22]) was not visible on the recorded thermograms since the temperature range used was of −40 °C to 100 °C), and (*ii*). the measured values corresponded to the Tg of the PMLABe block having an organization disturbed by the PEG block thus leading to a sharp reduction in the Tg of this PMLABe block. The second hypothesis that can be formulated was based on a phenomenon described by Bae et al. [23] on PEG-*b*-poly(L-lactic acid) block copolymers. These authors observed: (*i*). a single large peak of Tg in a wide temperature range and an absence of Tg from the amorphous zones of PLLA due to the phenomenon of high phase mixing between the PEG and PLLA blocks, and (*ii*). a significant decrease in temperature crystallization of PEG due to the presence of the PLLA block [23]. Based on this work, it can be assumed that the two blocks, PEG and PMLABe, constituting the copolymers considered in the present study might have a high degree of phase mixing resulting in: (*i*). the presence of a single glass transition temperature, and (*ii*). the absence of crystallization temperature of the PEG block. Further studies will have to be carried out to evidence, or not, the phase mixing between the two blocks. Nevertheless, the synthesized (co)polymers have the required properties allowing NPs formulation.

Several methods have been described in the literature for the conjugation of targeting agents, such as peptide or antibodies after formation of the NPs [14]. Among them, carbodiimide, maleimide and click chemistries are particularly prominent, each with their own advantages and drawbacks. Due to its ease of use, we chose to use maleimide-based Michael addition [17]. It also enabled us to compare the results obtained here with the ones previously obtained with the pre-formulation procedure, in which the thiol-terminated peptides were grafted on the polymer chain through Michael addition [12]. Indeed, when dealing with peptides, choice of the linker is not trivial and without consequences, since even the slightest modification can alter the peptide’s behavior [24,25]. It has been notably recently highlighted with radiolabeled peptide derivatives, for which the chelate conjugated to the peptide demonstrated a significant influence [26,27]. There even is a report of a somatostatin receptor antagonist peptide that unexpectedly turned into an agonist when coupled to a DOTA chelator [28].

Unmodified NPs, without or with maleimide groups at their surface, were successfully prepared by nanoprecipitation of corresponding mixture of PMLABe_73_ derivatives. Since their hydrodynamic diameters were lower than 100 nm, as already observed for PMLABe_73_-based NPs [12], with dispersity lower than 0.25, it can be concluded that quite narrow-dispersed NPs’ suspensions were obtained. Finally, the presence of 10 wt% of either maleimide or Mal-PEG_62_ groups on NPs’ surface had no significant influence on NPs’ characteristics.

Treatment of these native NPs under the conditions used to graft peptides (Michael addition conditions) showed that these conditions had no significant impact on NPs in terms of diameter and dispersity. Indeed, no drastic changes have been observed either for the diameter or dispersity values, since only non-significant diameter’s decrease (NPs 1′, NPs 7′ and NPs 8′—Table 1) or increase (NPs 2′—Table 1) were observed, while the dispersity values were nearly identical.

We then confirmed that peptides containing a thiol residue at their C-terminal end could be engrafted onto PMLABe-based NPs bearing maleimide groups by the Michael addition. Indeed, results highlighted that the grafting of peptides onto maleimide-decorated NPs, with or without a PEG spacer, did not destabilized the preformed polymeric NPs and that well-defined peptide-decorated nanoobjects were obtained. Those results have been compared to the ones we previously obtained [12] for peptide-decorated NPs obtained by nanoprecipitation of PMLABe_73_/Pept-PMLABe_73_ and PMLABe_73_/Pept-PEG_62_-*b*-PMLABe_73_ (90/10 wt%) mixture (Table 2).

The comparison between all the results (Appendix A) showed that native NPs (NPs 1′, NPs 2′, NPs 7′ and NPs 8′ Table 1; and NPs 1 and NPs 7, Figure 2) had similar characteristics in terms of diameter (Appendix A) and dispersity (Appendix A). On the other hand, NPs decorated post-formulation with GBVA10-9 and GBVA10-9scr (NPs 3, NPs 4, NPs 9 and NPs 10—Table 1) had significantly higher diameters (Appendix A) than corresponding NPs obtained by nanoprecipitation of peptide-functionalized (co)polymers (NPs 15, NPs 16, NPs 20 and NPs 21—Table 2). A significant increase in dispersity (Appendix A) has been also observed between NPs obtained by the post-formulation procedure vs. NPs obtained through the pre-formulation method. Such results most probably indicate that the GBVA10-9 and GBVA10-9scr were effectively located at the surface of NPs modified after their formulation thus increasing their diameters, while they were entrapped into the hydrophobic inner core of NPs prepared from peptide-functionalized (co)polymers. For the CPB and CPBscr-decorated NPs, we have observed a slight decrease in the values of diameters (Appendix A) for the NPs decorated with these peptides after their formulation (NPs 5, NPs 6, NPs 11 and NPs 12—Table 1) in comparison to the ones obtained from peptide-functionalized (co)polymers (NPs 17, NPs 18, NPs 22 and NPs 23—Table 2). An increase in the dispersity values for the CPB- and CPBscr-decorated NPs obtained by modification of the formulated NPs has been observed (Appendix A). The slight decrease in the diameters values between the two formulation methods might be due to the more hydrophilic character of CPB and CPBscr, leading to a more expensed conformation of these peptides in aqueous media. Finally, TEM images of peptide-modified NPs prepared by the post-formulation procedure (Figure 4) and by the pre-formulation procedure [12] were similar, thus highlighting that the procedure used to graft peptides had no influence on the NPs morphology.

Then, we looked at the impact of the functionalization by two peptides, GBVA10-9 and CPB, previously shown to have a strong tropism for hepatoma cells [11], on the NP’s cell internalization. We also used scrambled peptides for both CPB and GBVA10-9 to investigate whether their specific amino acid sequence was the main determinant to drive the hepatoma cell uptake of NPs functionalized by peptides with a post-formulation method (Figure 5). Our data showed that the peptide-functionalized NPs with both amphipathic-helical GBVA10-9 and scrambled peptide strongly enhanced internalization by hepatoma cells. Significant differences in the cell uptake between NPs functionalized with peptides using the Michael addition, in post-formulation, and NPs prepared from peptide-modified PMLABe (co)polymers have been observed (Figure 5 and Figure 6).

For instance, we previously reported that cell uptake of NPs prepared from GBVA10-9 functionalized PMLABe_73_ homopolymer was very low (Figure 6) [12] while, in this report, we showed that addition of the same peptide onto maleimide modified NPs strongly enhanced their cell uptake (Figure 5). Similarly, CPB- and CPBscr-modified NPs by post-formulation were internalized nearly at the same levels (Figure 5) while the uptake of NPs prepared from CPB-modified PMLABe_73_ homopolymer in pre-formulation was much higher than that of CPBscr-functionalized NPs (Figure 6). On their side, Valetti et al. found that conjugation of their peptide prior NPs formulation led to a better cellular uptake [13].

These data demonstrated that the method used to functionalize NPs with peptides, post- vs. pre-formulation, strongly influence the cell uptake of the nanovectors, and cannot be predicted *a priori*. Since there was no obvious relationship between neither the NPs’ diameters nor the NPs’ dispersity values and the uptake of such NPs, these features could not explain the differences in cell internalization. Moreover, all NPs obtained by pre- or post-formulation had spherical shape, which could not provide a clear explanation for the differences in NP’s cell uptake.

It might be postulated that the peptide grafting method, post- vs. pre-formulation, had a significant influence on the peptide accessibility and cell recognition capacity. In addition, we previously demonstrated that the amphipathic-helical GBVA10-9 peptide [29] could interact with apolipoproteins present in the culture medium, which strongly affected the opsonization of the NPs functionalized with this peptide [11]. The functionalization methods could also affect the binding of plasma proteins that would explain some of the differences observed in the cell internalization of peptide-decorated NPs.

Together, our data confirmed a previous hypothesis that peptides physicochemical features, the method used to functionalize NPs and/or formation of a protein corona could be the main drivers of the NPs’ cell internalization rather than the recognition of a membrane receptor mediated by a specific amino acid sequence of the peptides. Rigorous investigations on all these aspects are warranted in order to fully understand their uptake mechanism and how the different moieties interact, aiming at the most efficient tumor targeting.

## Figures and Tables

**Figure 3 polymers-14-02447-f003:**
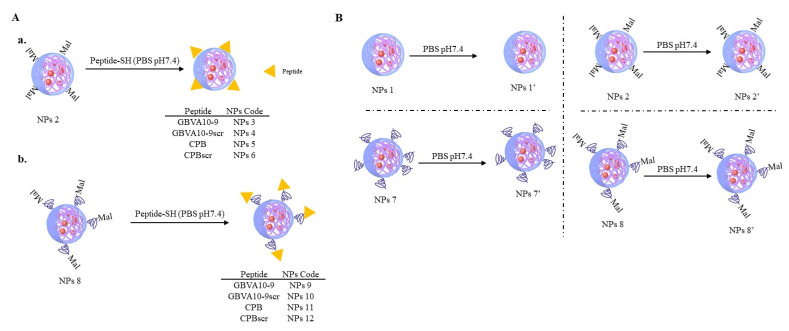
(**A**). Chemical grafting of peptide on: (**a**) Maleimide-decorated PMLABe_73_/Mal-PMLABe_73_-based NPs 2, and (**b**) Maleimide-decorated PMLABe_73_/Mal-PEG_62_-*b*-PMLABe_73_-based NPs 8; (**B**) Incubation of native (NPs 1 and NPs 7) and maleimide-decorated NPs (NPs 2 and NPs 8) under Michael reaction conditions to produce NPs 1′, NPs 2′, NPs 7′ and NPs 8′.

**Figure 4 polymers-14-02447-f004:**
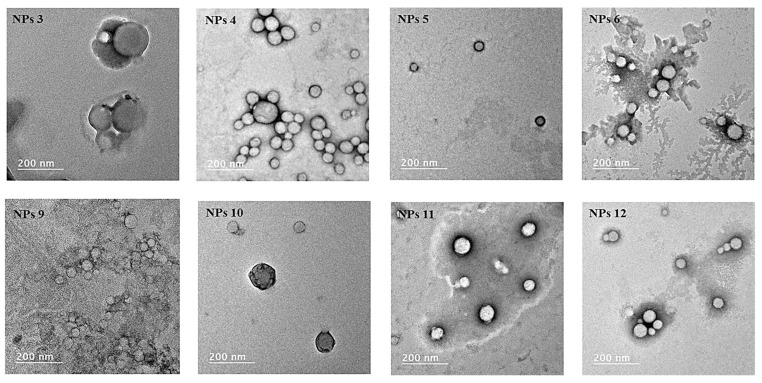
TEM images of peptide-decorated NPs obtained through post-formulation peptides grafting onto maleimide-decorated NPs.

**Figure 5 polymers-14-02447-f005:**
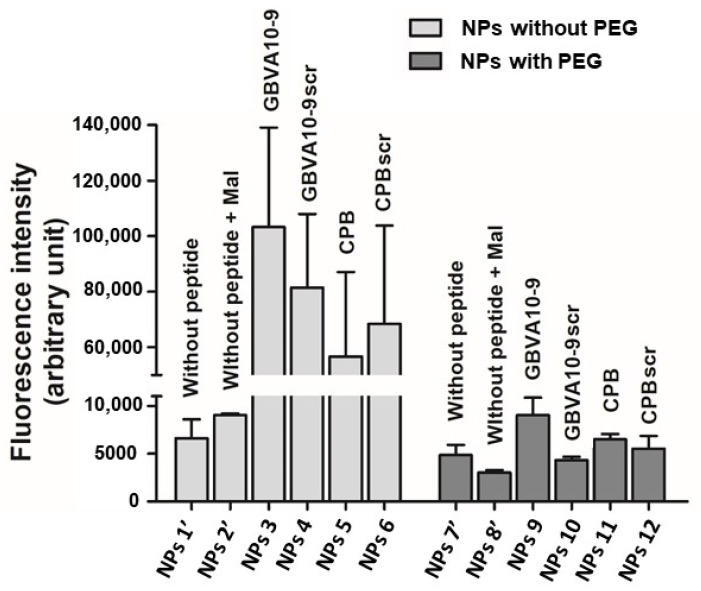
In vitro nanoparticle’s uptake assays using progenitor HepaRG cells of peptide-decorated NPs obtained by the post-formulation method.

**Figure 6 polymers-14-02447-f006:**
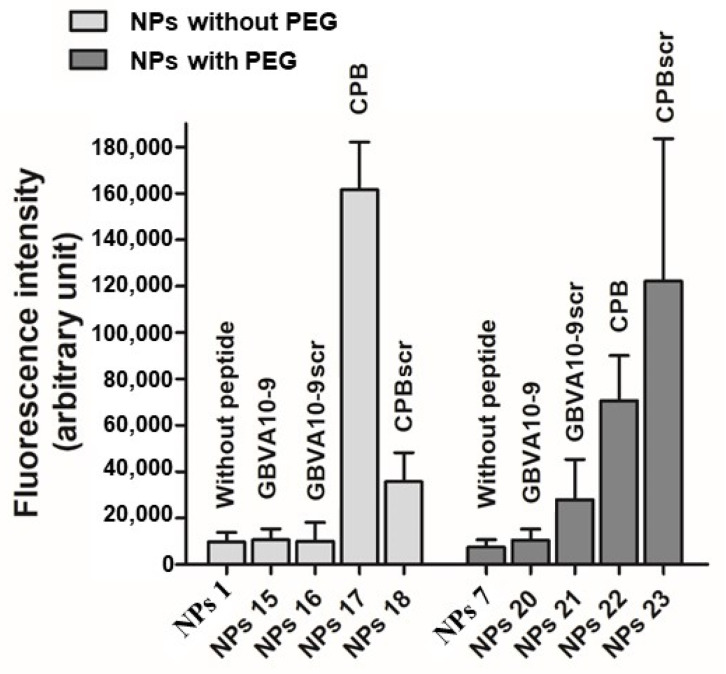
In vitro nanoparticle’s uptake assays using progenitor HepaRG cells of peptide-decorated NPs obtained by the pre-formulation method (Adapted from ref [12]).

**Table 2 polymers-14-02447-t002:** Composition of NPs obtained by nanoprecipitation of peptide-functionalized PMLABe_73_ derivatives [12].

Nature	Code	Composition	Dh (nm) ^a^	PDI ^a^
Peptide-decorated NPs	NPs 15	PMLABe_73_/GBVA10-9-PMLABe_73_ (90/10)	108	0.18
	NPs 16	PMLABe_73_/GBVA10-9scr-PMLABe_73_ (90/10)	115	0.16
	NPs 17	PMLABe_73_/CPB-PMLABe_73_ (90/10)	139	0.12
	NPs 18	PMLABe_73_/CPBscr-PMLABe_73_ (90/10)	163	0.10
Peptide-decorated PEGylated NPs	NPs 20	PMLABe_73_/GBVA10-9-PEG_62_-*b*-PMLABe_73_ (90/10)	66	0.21
	NPs 21	PMLABe_73_/GBVA10-9scr-PEG_62_-*b*-PMLABe_73_ (90/10)	106	0.17
	NPs 22	PMLABe_73_/CPB-PEG_62_-*b*-PMLABe_73_ (90/10)	128	0.13
	NPs 23	PMLABe_73_/CPBscr-PEG_62_-*b*-PMLABe_73_ (90/10)	146	0.13

^a^ Dh and PDI were measured by DLS (3 runs of 70 scans each).

## Data Availability

Data presented in the results are not publicly available but Appendix A can be downloaded.

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
