# Peer review of "Hepatotropic Peptides Grafted onto Maleimide-Decorated Nanoparticles: Preparation, Characterization and In Vitro Uptake by Human HepaRG Hepatoma Cells"

_polymers, 2022, doi:10.3390/polym14122447_

Round 1
Reviewer 1 Report
The manuscript entitled “Hepatotropic peptides grafted onto maleimide-decorated nanoparticles: preparation, characterization and in vitro uptake by human HepaRG hepatoma cells” by Brossard et al. reports a methodology for synthesis by precipitation and functionalization of the polymeric particles. An important effort in the synthesis of the nanoparticles is possible to appreciate. After some characterizations, the authors evaluate the performance of the nanomaterials for internalization in hepatic cells. However, from the point of view of the polymeric materials characterization, the manuscript presents some weaknesses. In this regard, I suggest that the authors complete the characterization of the systems herein proposed to reach a better match with Polymers journal. Essential comments are attached below:
1- Abstract. I recommend improving the organization and redaction of this part. Please, consider the inclusión of some numerical data that supports and summarizes your findings.
2- Introduction. The background and relevance of the problem are not clear. Improve it.
3- Did you measure the superficial charge for the nanoparticles? It would be an interesting characterization of the systems presented before and after the functionalization of the particles.
4- Regarding SEC results… why in some graphs the detector response is positive but in others is negative?
5- Methods. NPs formulation and peptides’ grafting procedure:. Please, organize in order to be clearer. The current form is not adequate.
6- Table 1 and 2. Results are average? DLS was measured and reported by Intensity? Please, show the distribution curve. Besides, it should be Informed the number of runs and SD for hydrodynamic diameters and PDI.
7- Is redundant the inclusión of Table 3 and Figure 4.
8- The characterization of the systems is so poor from a polymeric and nanomaterials point of view. Please, improve this point, considering Surface charge, thermal analysis, stability and degradation behavior of the particles, and spectroscopies studies.
9- Are the nanoparticles capable of being swelled? Please, consider this fact.
10- Section 3.2. Nanoparticle’s uptake by HepaRG hepatoma cells in vitro. Results should be discussed more deeply. Is possible to complement other studies?
Reviewer 2 Report
In this manuscript the authors describe the development and functionalization of nanoparticles for the treatment of HCC. It is a complete and interesting experimental work. However, I consider that the discussion of the results should be improved, as it is very scarce in the current manuscript.
Reviewer 3 Report
It was a manuscript about synthesis, characterization, and cell uptake evaluation different nano formulation to detect the effect of functionalization method on nanoparticles’ cellular uptake. Here are some comments on this study that should be considered before publication:
- The quality of abstract is not good. Please improve it.
- There are grammatical mistakes in the text that should be corrected.
- Please add the manufacturer country of apparatuses.
- “Native and maleimide-decorated NPs, without (NPs 1 and NPs 2, respectively) or with 10 wt% of PEG (NPs 7 and NPs 8, respectively),” this sentence needs to be rewritten.
- “For the CPB and CPBscr-decorated NPs, we have observed a slight decrease in the values of diameters (Figure 4A) for the NPs decorated with these peptides after their for mulation (NPs 5, NPs 6, NPs 11 and NPs 12 – Table 2) in comparison to the ones obtained from peptide-functionalized (co)polymers (NPs 17, NPs 18, NPs 22 and NPs 23 – Table 3), varying from 130-160 nm for the latter ones to 80-100 nm for the former ones. An increase in the dispersity values for the CPB- and CPBscr-decorated NPs obtained by modification of the formulated NPs has been observed (Figure 4B). The slight decrease in the diameters values between the two formulation methods might be due to the more hydrophilic character of CPB and CPBscr, leading to a more expensed conformation of these peptides in aqueous media.” this paragraph is written twice.
- The discussion part of the paper is poor. Please explain more about your results and what you finally obtained from each part.
- It is not normal to use references in conclusion part.
- Please also improve the quality of abstract.
- What is the difference between NP 14 with 1, and NP 19 with 7?
- “. Peptide grafting led to an increase in 81 NPs’ dispersity and either an increase or a decrease in NPs’ diameter, more or less important in function of the considered peptide and/or (co)polymers, while TEM images demonstrated that the modified nanoparticles kept their spherical morphology. Then, peptide-modified NPs produced by post-formulation method and encapsulating the fluorescent probe DiD Oil were incubated with HepaRG hepatoma cells in order to evaluate their internalization. Results were compared to the ones we have previously obtained with peptide-decorated NPs produced by the pre-formulation method [12]. Flow cytometry analysis showed that native and maleimide-decorated NPs were internalized by Hep-aRG hepatoma cells in low amounts, while the uptake of peptide-decorated NPs prepared from PMLABe73/Mal-PMLABe73 was much stronger and relied on the peptide’s nature. In addition, significant differences in NP’s uptake were observed between NPs prepared using the pre- and post-formulation methods.” these are results. Please don’t mention them in introduction section.
Round 2
Reviewer 1 Report
I understand is not possible to evaluate surface charge. However, I consider at least some characterization techniques proper for polymers are required if you want to publish in this Journal (POLYMERS). Regarding the swelling behavior, I did not get how are authors thinking to use the reported particles, could the authors explain this point better?. In general, the other questions were solved.
Reviewer 3 Report
Thanks for addressing the comments.
Author Response
See attached letter

Round 3
Reviewer 1 Report
The manuscript was improved, being now acceptable for its publication.